# Effectiveness of health education interventions for cervical cancer screening: A quasi-experimental study in Pokhara Metropolitan Slum, Nepal

Abhishek Sapkota ⓘ*, Chiranjivi Adhikari ⓘ*, Hari Prasad Kaphle ⓘ,
Birkha Bahadur Bist ⓘ

School of Health and Allied Sciences, Pokhara University, Pokhara, Kaski, Nepal

* abhishek.sapkota2153@gmail.com (AS); chiran.iom13@gmail.com (CA)

## Abstract

Cervical cancer remains a major public health challenge in low- and middle-income countries, including Nepal, particularly among marginalized populations such as women living in slum areas. Despite national screening guidelines and awareness campaigns, uptake of cervical cancer screening remains low. The study aimed to assess the effectiveness of a community information channel (CIC) and community health worker (CHW)-led interventions on improving the knowledge and utilization of cervical cancer screening among women of slum areas in Pokhara metropolitan, Nepal. A quasi-experimental study with a pre-test and post-test control group design was conducted in two purposively selected slum areas. Women aged 30–60 years (n = 310) were recruited and divided into intervention and control groups. The intervention group received CHW-led individual health education using flipcharts and brochures, while the control group was exposed to CIC-led awareness through posters. Data were collected at baseline, midterm (3rd week), and endline (6th week) using structured questionnaires and analyzed using SPSS (version 22) and R, where appropriate statistical tests were carried out. Knowledge scores increased significantly in both groups; however, the intervention group showed a greater magnitude of change, with the median knowledge score rising from 2 to 8 compared to an increase from 3 to 5 in the control group. Additionally, screening uptake in the intervention group rose significantly from 36.1% to 52.8% (p-value < 0.001). Improvements were also observed in perceived susceptibility (median difference = 1, 95% CI: 0–2; p < 0.001), perceived severity (median difference = 1, 95% CI: 0–1; p = 0.0011), and behavioral intention (median difference = 15, 95% CI: 0–60; p < 0.001) post-intervention. Community health worker (CHW)-led intervention was effective in enhancing knowledge, perceived susceptibility, perceived severity, behavioral intention, and thereby increasing cervical cancer screening uptake.

**Data availability statement:** The data used for the analysis in this study is included in the supporting information file.

**Funding:** "This study was partially funded by the Gandaki Province Academy of Science and Technology, Pokhara, Nepal (Grant number: Gpast24_07 to AS, NPR 78900). The funder had no role in study design, data collection and analysis, decision to publish, or preparation of the manuscript. No additional external funding was received for this study".

**Competing interests:** The authors have declared that no competing interests exist.

**Abbreviations:** CC: Cervical Cancer; CCS: Cervical Cancer Screening; CHW: Community Health Worker; CIC: Community Information Channel; HBM: Health Belief Model; SI: Screening Intention; SU: Screening Uptake; TPB: Theory of Planned Behavior; VIA: Visual Inspection with Acetic Acid; WHO: World Health Organization.

## Introduction

Cervical cancer ranks as the fourth most prevalent cancer among women worldwide, with approximately 660,000 new diagnoses reported in 2022 [1]. In the same year, nearly 94% of the 350,000 cervical cancer-related deaths occurred in low- and middle-income countries [1]. The regions with the highest incidence and mortality rates of cervical cancer include sub-Saharan Africa, Central America, and Southeast Asia [1–3]. The disparities in cervical cancer prevalence across regions are attributed to inequalities in access to resources for prevention, screening, and treatment of cervical cancer, as well as social and economic factors such as poverty and gender biases [1,4]. Cervical cancer predominantly affects younger women, with 20% of children losing their mothers to this disease [5].

Nepal, a South Asian nation, experiences an unusually high incidence of cervical cancer, underscoring the disparities associated with this disease. In 2020, cervical cancer continued to be the leading cancer diagnosed among Nepalese women, resulting in the highest number of cancer-related deaths [6]. The age-standardized incidence rate stood at 16.4 cases per 100,000 women, with 2,244 new diagnoses and 1,493 fatalities [6]. The cervical cancer screening and prevention guidelines were established in Nepal in 20 [10] [7]. According to these national guidelines, women aged 30–60 are advised to undergo visual inspection with acetic acid (VIA) screening every five years, which initially targeted screening at least 50% of women aged 30–60 years and later revised to 70% in 2020 [7]. However, by 2019, only 8.2% of women aged between 30 and 49 had undergone screening [8]. Several obstacles, including limited awareness, limited trained healthcare professionals, and feelings of embarrassment, hinder the utilization of cervical cancer screening [9]. On the other hand, factors like the gender of healthcare providers, effective counseling, and the privacy of screening services are frequently cited as facilitators for screening [8].

Socio-demographic factors affect perceived risk, access to services, and social support, which are key determinants in HBM [11]. Behavioral factors influence perceptions of vulnerability and social norms, which affect health-related decision-making [12]. Knowledge of cervical cancer affects beliefs about perceived susceptibility, perceived severity, and behavioral intention for seeking screening [13].

Community Information Channel-led interventions using posters are provided by the health facilities to aware women of cervical cancer [7]. According to the 2nd amendment of the National Female Community Health Volunteer Program Strategy in 2076 BS, FCHVs must have at least passed the School Leaving Certificate (SLC) examination [14]. The fragmented nature of health programs poses a challenge for Female Community Health Volunteers in coordinating activities and delivering results [15]. So, an integrated CHW program model is needed to support the implementation of community-based health interventions for the prevention and control of cervical cancer effectively.

Women in slum areas may have limited knowledge about cervical cancer, its risk factors, prevention methods, and the importance of early detection through screening. This lack of awareness can lead to delayed diagnosis and treatment, exacerbating the disease burden. In the meantime, as the Nepal Government [7] and the WHO

[16] proposed the 90-70-90 targets for 2030, aiming to eliminate cervical cancer (CC), one of which targets was to screen 70% of women aged 30–60 years for CC. For this, the Nepal Government may strategize as learning from the community information channel (CIC) such as street banners at the outset, and then involving the community health workers (CHWs) are highly effective in improving knowledge, attitude and increasing the cervical cancer screening seeking behaviors from 12 percent point to 61 percent [17]. Similarly, a systematic review showed that lecture-only or lecture-demonstration with video interventions in communities can significantly improve knowledge and awareness [18]. However, there was a need to evaluate the effectiveness of such interventions in specific settings, such as slum areas in Pokhara, Nepal.

The objective of this study was to assess the effectiveness of community information channel (CIC) and community health worker (CHW)-led interventions on improving the cervical cancer knowledge and screening uptake among women of slum areas in Pokhara metropolitan, Nepal. The findings could guide the development and implementation of effective interventions to reduce the burden of cervical cancer and improve women's health outcomes in similar settings globally.

## Materials and methods

### Study design and setting

The study conducted in two slum areas of Pokhara Metropolitan, Nepal, between 29th January 2025 and 13th March 2025 implemented a Quasi-experimental (pre-test-post-test control group) study design involving two groups: intervention (community health worker-led using a flipchart and brochures) and control (community information channel-led using a poster) Fig 1.

### Study participants and sampling procedure

Women aged 30–60 years from two slum areas of Pokhara Metropolitan City were recruited as study participants. The inclusion criteria for the study participants were women aged 30–60 years residing in the slum area, those who have not been diagnosed with cervical cancer, and those who are not planning to migrate to another place in the next 3 months. The exclusion criteria were women who are currently pregnant and those who have undergone a hysterectomy. In the first stage, two slum areas of the Pokhara Metropolitan City were selected purposively to ensure maximum matching between both slum areas, to ensure participants from both the control and intervention groups share similar characteristics. To control the potential confounding effect of accessibility on screening uptake, slum areas with similar distances to health facilities with VIA services were selected. In the second stage, one slum area was conveniently selected as the intervention group and the other one was conveniently designated as the control group. In the third stage, the list of women aged 30–60 years in both the selected slum areas were prepared with the help of the head of the slum areas, and the participants were randomly enrolled in the study.

### Sample size calculation

The sample size was calculated using screening rate at baseline study = 73% (p1), the expected change in proportion up to 87% (p2), [19] with other values conventionally, as $Z_{a/2} = 1.96$ (for 95% CL), $Z_{\beta} = 0.84$ ($\beta = 80\%$), and using the formula for sample size (n)=[$(Z_{a/2} + Z_{\beta})^2$. (p1(1-p1)+p2 (1-p2))]/(p1-p2)^2, we obtained 124.08 for each group. With an attrition rate of 20%, the final sample size for each group was 155.

### Operational definitions

**Cervical Cancer Screening (CCS) Uptake:** Participants who have undergone cervical cancer screening in the last 5 years.
**Community Health Worker (CHW):** The CHW refers to the participants who have passed the SLC and were trained to provide awareness in the intervention group [20].
**CHW-led intervention:** CHW-led intervention refers to the intervention provided by trained CHWs in the intervention group using flip charts and brochures to inform women about slum areas.

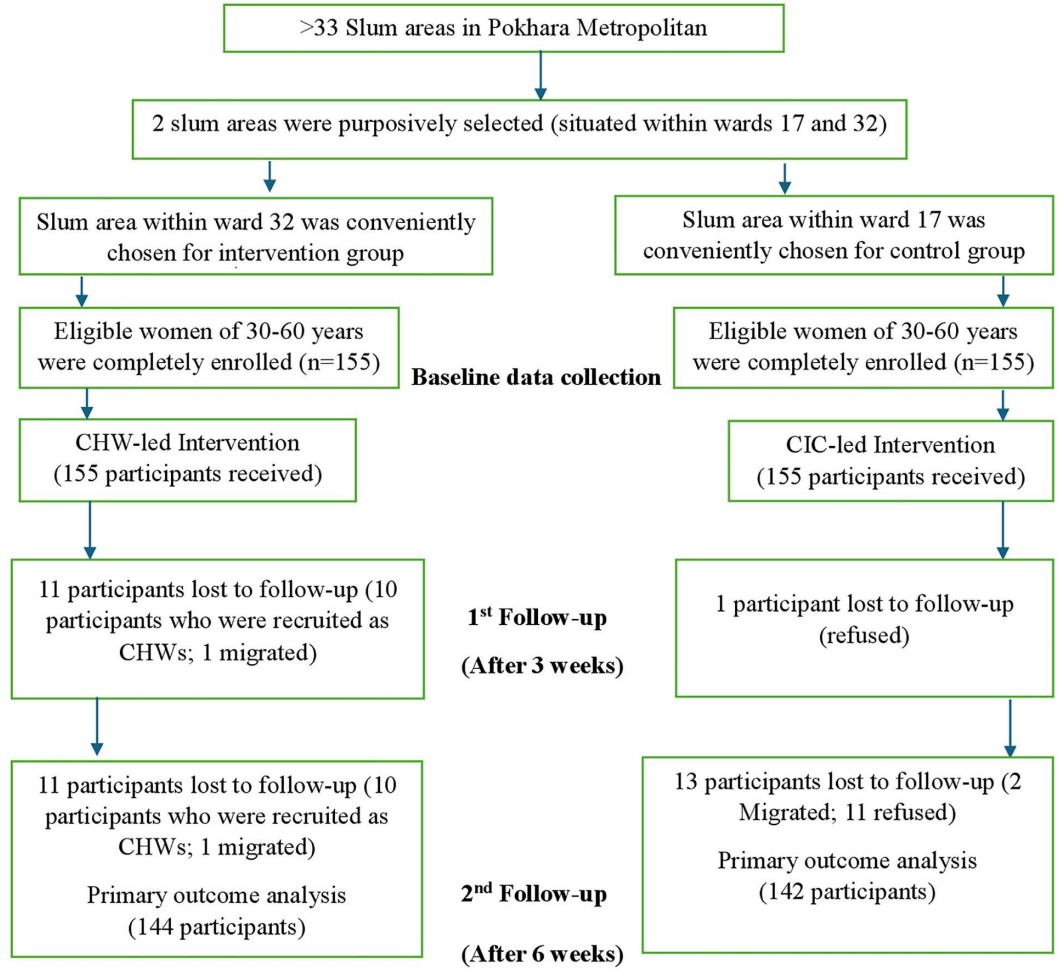

**Fig 1. CONSORT flow chart.**

**CIC-led intervention:** CIC-led intervention refers to the use of posters to raise awareness of the prevention and control of cervical cancer in the control group.

## Data collection

Data was collected by trained female data enumerators from a public health background through face-to-face interviews using a structured questionnaire adapted from relevant literature and previous studies on similar topics. Pre-testing of the study tool was done among 31 women aged 30–60 years in a slum area within ward 29 of Pokhara Metropolitan to ensure it reliability and validity.

A baseline study was conducted in the selected slum areas to assess the knowledge of cervical cancer and demographic and behavioral characteristics of the participants. Post-test was done among both the study groups (intervention and control) after the third week and the sixth week. Data were monitored by the principal investigator and the supervisor.

## Development and testing of intervention packages

A thorough review of existing literature on health education interventions for the prevention and control of cervical cancer. Health education material-the flip chart was designed to be highly pictorial. The content was developed in a culturally and

linguistically appropriate manner. While brochures and posters were adopted from the website of the National Health Education, Information, and Communication Centre (NHEICC), Nepal. Contents of the health materials covered introduction, risk factors, symptoms, prevention and control measures, and screening tests for cervical cancer.

The training materials and intervention packages were pilot-tested in a small group in the slum areas of Ward 29 to evaluate feasibility and acceptability. Five women were trained using the presentation slides, and all scored above 90% on the post-training assessment, demonstrating adequate knowledge acquisition. These trained CHWs were then mobilized in their local areas to provide health education to two eligible participants each, using flip charts and brochures. Significant improvements in participants' knowledge were observed after receiving the education sessions, indicating the effectiveness of the CHW-led intervention. The effectiveness of the posters was not tested during the pilot, as they were adopted directly from the National Health Education, Information and Communication Centre (NHEICC), Nepal, and are already widely used in health facilities. Feedback from both CHWs and participants was used to refine the intervention package, which was then finalized for intervention implementation.

### CHW training

Ten participants who had at least passed the SLC were enrolled as community health workers (CHWs) in the intervention group after the completion of the baseline study. Training on cervical cancer was provided to the CHWs by the principal investigator and a nurse using presentation slides. The training was for one day (4 hours). Each CHW was assigned 15–16 participants to deliver health education.

### Interventions

**CHW-led intervention.** After receiving one day of training, community health workers (CHWs) visited the homes of every participant in the intervention group to provide a single health education session on cervical cancer. Using standardized flip charts, CHWs delivered information on cervical cancer risk factors, early warning signs, prevention and control measures, the importance of screening, available screening methods, and the benefits of early detection and treatment. Each session lasted approximately 15 minutes and was designed to be interactive, allowing participants to ask questions. CHWs also distributed brochures summarizing the key messages. The intervention was monitored by the principal investigator to ensure fidelity. All participants in the intervention group received this education session before the start of the midterm survey.

**CIC-led intervention.** In the control group, a minimal CIC-led intervention was provided to uphold the rights of study participants. Specifically, after the baseline survey, informational posters focusing on cervical cancer prevention and screening were displayed at two different locations within the study site. The posters contained both pictorial and descriptive information in Nepali. The posters remained in place throughout the study period and continued to be displayed even after the completion of the study.

The CHW-led intervention required substantially greater human and material resources compared to the CIC-led intervention, which consisted only of a poster display.

### Outcome measures

The primary outcome was the change in knowledge of cervical cancer. Secondary outcomes included changes in cervical cancer screening uptake, perceived susceptibility, perceived severity, and behavioral intention related to cervical cancer from baseline to midterm and endline assessments.

Knowledge was measured using 14 structured questions (maximum score: 36) covering cervical cancer introduction, risk factors, symptoms, screening methods, and prevention and control measures. Higher scores indicated greater knowledge. Screening uptake was measured by asking whether the participant had undergone cervical cancer screening within the past five years; those who had were classified as having screening uptake. Perceived susceptibility was assessed

with a single-item question: *"How likely do you think it is that you could develop cervical cancer?"* Responses were rated on a 5-point Likert scale (1 = very unlikely, 5 = very likely), with higher scores indicating greater perceived susceptibility. Perceived severity was measured with the question: *"If you were to get cervical cancer, how much of an impact would it have on your life?"* Responses were rated on a 5-point Likert scale (1 = no impact, 5 = extreme impact), with higher scores reflecting greater perceived severity. Behavioral intention was assessed by asking participants to estimate the likelihood of undergoing cervical cancer screening in the future, rated on a probability scale from 0% (not at all likely) to 100% (certain).

## Data analysis

All quantitative data were analyzed using the Statistical Package for Social Sciences (SPSS, version 22). Descriptive statistics were used to summarize baseline characteristics: medians, minimum and maximum for continuous variables, and frequencies with percentages for categorical variables.

For outcome analysis, appropriate statistical tests were applied based on the type of variable and study design. Changes in knowledge scores (continuous, non-normally distributed) from baseline to midline and endline were assessed using the Wilcoxon signed-rank test. Perceived susceptibility and perceived severity (ordinal, 5-point Likert scale) were analyzed similarly. Behavioral intention (0–100% probability) was analyzed as a continuous variable using a non-parametric test (Wilcoxon signed-rank). Cervical cancer screening uptake (binary, yes/no) was compared within groups using McNemar's test and between groups using Pearson's chi-square test. For between-group chi-square tests of a 2 × 2 table, the effect size was reported as Phi ($\varphi$).

For outcomes that were not approximately normally distributed, we reported medians and ranges (minimum–maximum). Group differences were assessed using the median difference, with 95% confidence intervals obtained via non-parametric bootstrap resampling in R (Version: RStudio/2025.05.1 + 513). To assess the overall trend over time, each subject's scores across the three time points were averaged to calculate an overall median for knowledge, perceived susceptibility, perceived severity, and behavioral intention. No interim analysis was performed due to the short time frame.

## Benefits to participants and handling of possible risks

The study was done in such a way that no harmful effects were created for the participants. The participants benefited from the health educational intervention, which focused on cervical cancer screening behavior and knowledge related to the prevention and control of cervical cancer.

## Ethical considerations

The ethical approval was taken from the Institutional Review Committee (IRC), Pokhara University (Ref no. 169/2081/82). Written consent from the participants was obtained before the study began. Data collection was conducted in the Nepali language to ensure participants' understanding. Withdrawal was accepted at any time during data collection. Anonymity was maintained by keeping code numbers in questionnaires. Confidentiality was maintained by using the obtained information for the study only.

## Dissemination plan

Data will be presented at national and international conferences.

## Trial registration

The study trial was registered in the Australian New Zealand Clinical Trials Registry (ANZCTR). The trial registration number is ACTRN12625000497404. https://anzctr.org.au/ACTRN12625000497404.aspx

## Results

A total of 310 (100%) women aged 30–60 years participated in the baseline study, with 155 each from the intervention and control slum areas. The response rates in the midterm study were 92.9% and 99.4% in the intervention (144) and control groups (154), respectively. Moving on to the endline study, the response rate was 92.9% and 91.6% in the intervention (144) and control groups (142), respectively.

### Baseline characteristics

The median (min-max) age in years of the participants from the intervention and control groups was 39 (30–60) years and 41 (30–59) years, respectively. Almost half (49.7%) of the participants from the intervention group belonged to the age group (30–40) years, while almost two-fifths (39.4%) of the participants from the control group belonged to the age groups (30–40) years and (40–50) years. The majority of the participants from both groups had a basic level of education, i.e., 45.2% and 49% in the intervention and control groups. More than half (51%) of the participants in the intervention group were Dalits, while the majority of the participants (34.8%) of control group belonged to the Relatively Advantaged Janajati ethnic group. Most of the participants from both groups were married. More than four-fifths of the participants in the groups were housewives. The median income in Nepalese rupees was 20000 and 30000 in the intervention and control groups, respectively. More than half (60%) of the participants in the control group and a minority (46.5%) of the participants in the intervention group had a monthly income greater than or equal to 25000. The proportion of current smokers was 12.3% and 6.5% in the intervention and control groups. The major source for health information in both groups was from the Female Community Health Volunteer. Almost all (99.4%) of the participants from both groups had a single sexual partner. The proportion of family history of cervical cancer was also the same in both groups, with 6.5%. The baseline characteristics were generally comparable between the intervention and control groups, except for the significant differences in ethnicity, monthly income, and major source for health information (p-value <0.05) (Table 1).

### Cervical cancer-related knowledge, perceptions, and intention between groups over time

At baseline, there were no statistically significant differences between the intervention and control groups in perceived susceptibility (p=0.68; median difference=0, 95% CI: –1–1), perceived severity (p=0.22; median difference=0, 95% CI: 0–0), or behavioral intention (p=0.50; median difference=0, 95% CI: –30–20). A small but statistically significant difference was observed for knowledge (p=0.014; median difference=−1, 95% CI: –1–0).

At midterm, all outcomes showed statistically significant differences favoring the intervention group, including knowledge (p<0.001; median difference=2.5, 95% CI: 1 to 5.5), perceived susceptibility (p<0.001; median difference=1, 95% CI: 0–2), perceived severity (p=0.0011; median difference=1, 95% CI: 0–1), and behavioral intention (p<0.001; median difference=15, 95% CI: 0–60).

At endline, these differences remained statistically significant for knowledge (p<0.001; median difference=3, 95% CI: 1–5), perceived susceptibility (p<0.001; median difference=1, 95% CI: 0–1), perceived severity (p=0.029; median difference=1, 95% CI: 0–1), and behavioral intention (p<0.001; median difference=20, 95% CI: 0–65). (Table 2)

### Comparison of the overall median scores of cervical cancer-related knowledge, perceptions, and intention between groups

The intervention group had higher overall medians compared with the control group for knowledge (median=6; 95% CI: 5.00–7.33 vs median=4.33; 95% CI: 3.67–5.00; p=0.0067), perceived susceptibility (median=3; 95% CI: 3.00–3.33 vs median=2.67; 95% CI: 2.33–3.00; p<0.001), and behavioral intention (median=50; 95% CI: 50–63.33 vs median=36.67; 95% CI: 33.33–46.67; p<0.001). For perceived severity, the overall medians were the same (median=4.33), but the confidence intervals differed slightly (intervention 95% CI: 4.33–4.67 vs control 95% CI: 4.00–4.33), and the difference was

**Table 1. Baseline characteristics of the participants.**

| Variables | Intervention group | | Control group | | p-value |
|---|---|---|---|---|---|
| | n | % | n | % | |
| **Age group (years)** | | | | | |
| 30-40 | 77 | 49.7 | 61 | 39.4 | 0.11[m] |
| 40-50 | 46 | 29.7 | 61 | 39.4 | |
| 50-60 | 32 | 20.6 | 33 | 21.3 | |
| Median (Min-Max) | 39 (30-60) | | 41 (30-59) | | |
| **Level of education** | | | | | |
| Non-formal | 50 | 32.3 | 58 | 37.4 | 0.070[c] |
| Basic | 70 | 45.2 | 76 | 49.0 | |
| Secondary | 25 | 16.1 | 19 | 12.3 | |
| University or higher | 10 | 6.5 | 2 | 1.3 | |
| **Ethnicity** | | | | | |
| Dalit | 79 | 51.0 | 47 | 30.3 | **<0.001[c]** |
| Disadvantaged Janajati | 35 | 22.6 | 34 | 21.9 | |
| Relatively Advantaged Janajati | 23 | 14.8 | 54 | 34.8 | |
| Upper caste | 18 | 11.6 | 20 | 12.9 | |
| **Marital status** | | | | | |
| Single[a] | 14 | 9.0 | 15 | 9.7 | 0.85[c] |
| Married | 141 | 91.0 | 140 | 90.3 | |
| **Occupation** | | | | | |
| Housewife | 130 | 83.9 | 138 | 89.0 | 0.30[c] |
| Business | 13 | 8.4 | 11 | 7.1 | |
| Others[b] | 12 | 7.7 | 6 | 3.9 | |
| **Monthly income (in NPR)** | | | | | |
| <25000 | 83 | 53.5 | 62 | 40.0 | **0.0020[m]** |
| ≥25000 | 72 | 46.5 | 93 | 60.0 | |
| Median (Min-Max) | 20000 (1200-100000) | | 30000 (5000-80000) | | |
| **Smoking** | | | | | |
| Never smoker | 127 | 81.9 | 141 | 91.0 | 0.066[c] |
| Past smoker | 9 | 5.8 | 4 | 2.6 | |
| Current smoker | 19 | 12.3 | 10 | 6.5 | |
| **Major source for health information** | | | | | |
| Television | 6 | 3.9 | 14 | 9.0 | **0.014[c]** |
| Health worker | 50 | 32.3 | 37 | 23.9 | |
| FCHV | 75 | 48.4 | 63 | 40.6 | |
| Social media | 24 | 15.5 | 41 | 26.5 | |
| **Multiple sexual partners** | | | | | |
| No | 154 | 99.4 | 154 | 99.4 | 1.00[f] |
| Yes | 1 | 0.6 | 1 | 0.6 | |
| **Family history of cervical cancer** | | | | | |
| No | 145 | 93.5 | 145 | 93.5 | 1.00[c] |
| Yes | 10 | 6.5 | 10 | 6.5 | |
| **Total** | **155** | **100%** | **155** | **100%** | |

[a] = unmarried, separated, divorced, widow;

[b] = government job, non-government job, agriculture

[m] = Mann Whitney U test, [c] = Chi-square test, [f] = Fisher Exact test

**Table 2. Cervical cancer-related knowledge, perceptions, and intention between groups over time.**

| Time Point | Intervention | | Control | | p-value[m] | Median Difference | 95% CI |
|---|---|---|---|---|---|---|---|
| | n | Median (Min-Max) | n | Median (Min-Max) | | | |
| **Knowledge** | | | | | | | |
| Baseline | 155 | 2 (0-17) | 155 | 3 (0-14) | **0.014** | -1 | -1-0 |
| Midterm | 144 | 7.5 (0-30) | 154 | 5 (0-25) | **<0.001** | 2.5 | 1-5.5 |
| Endline | 144 | 8 (0-31) | 142 | 5 (1-25) | **<0.001** | 3 | 1-5 |
| **Perceived Susceptibility** | | | | | | | |
| Baseline | 155 | 3 (1-4) | 155 | 3 (1-5) | 0.68 | 0 | -1-1 |
| Midterm | 144 | 4 (1-5) | 154 | 3 (1-5) | **<0.001** | 1 | 0-2 |
| Endline | 144 | 4 (1-5) | 142 | 3 (1-5) | **<0.001** | 1 | 0-1 |
| **Perceived Severity** | | | | | | | |
| Baseline | 155 | 4 (1-5) | 155 | 4 (2-5) | 0.22 | 0 | 0-0 |
| Midterm | 144 | 5 (2-5) | 154 | 4 (2-5) | **0.0011** | 1 | 0-1 |
| Endline | 144 | 5 (3-5) | 142 | 4 (2-5) | **0.029** | 1 | 0-1 |
| **Behavioral Intention** | | | | | | | |
| Baseline | 155 | 40 (0-100) | 155 | 40 (0-100) | 0.50 | 0 | -30-20 |
| Midterm | 144 | 55 (0-100) | 154 | 40 (0-100) | **<0.001** | 15 | 0-60 |
| Endline | 144 | 60 (0-100) | 142 | 40 (0-100) | **<0.001** | 20 | 0-65 |

[m] = Mann-Whitney U test

statistically significant (p = 0.032). Overall, these results indicate a greater upward trend over time in the intervention group than in the control group. (Table 3)

## Cervical cancer screening uptake between the intervention and control groups over time

At baseline, cervical cancer screening uptake was similar between the control group (35.5%) and the intervention group (36.1%). At midterm, the uptake increased in both groups, with a higher proportion in the intervention group (47.9%) compared to the control group (37.0%). At endline, screening uptake further increased in the intervention group to 52.8%, while the control group remained relatively unchanged at 38.7%, which was statistically significant (p = 0.017, φ = 0.141).

Regarding the within-group comparison of cervical cancer screening, the intervention group showed statistically significant improvements in screening uptake from baseline to midterm (p < 0.001) and from baseline to endline (p < 0.001). Additionally, a modest but statistically significant increase was observed between midterm and endline (p = 0.016). (Table 4)

## Discussion

The community-based quasi-experimental study evaluated the effectiveness of community health workers and community information channels-led interventions on cervical cancer knowledge, perceived severity, perceived susceptibility, behavioral intention, and screening uptake among women aged 30–60 years living in slum areas of Pokhara Metropolitan, Nepal.

Both groups showed similar baseline characteristics except for source of information, income, and ethnicity distributions. Regarding changes in knowledge, perceived severity, and susceptibility, both (CHW-led and CIC-led) interventions showed statistically similar changes across three assessments; nonetheless, the overall trend over time was greater in the CHW-led intervention. More importantly, only the CHW-led intervention was found effective in bringing about the changes in screening intention (SI) and uptake (SU) behaviors. Moreover (and interestingly), SI was found changed in baseline-midterm, however, on the other hand, SU behavior was found progressively increasing baseline through endline. We discuss further from the light of ethnicity, income, and sources of information, which varied at baseline, including others.

**Table 3. Comparison of overall median scores of cervical cancer-related knowledge, perceptions, and intention between groups.**

| Variables | Intervention median (95% CI) | Control median (95% CI) | p-value[m] |
|---|---|---|---|
| **Knowledge** | 6.00 (5.00-7.33) | 4.33 (3.67-5.00) | **0.0067** |
| **Perceived susceptibility** | 3.00 (3.00-3.33) | 2.67 (2.33-3.00) | **<0.001** |
| **Perceived severity** | 4.33 (4.33-4.67) | 4.33 (4.00-4.33) | **0.032** |
| **Behavioral intention** | 50.00 (50.00-63.33) | 36.67 (33.33-46.67) | **<0.001** |

[m] = Mann-Whitney U test

**Table 4. Cervical cancer screening uptake between the intervention and control groups over time.**

| Groups | Cervical cancer screening uptake | | | | | | | | | | | |
|---|---|---|---|---|---|---|---|---|---|---|---|---|
| | Baseline | | | Midterm | | | Endline | | | Within-group comparison | | |
| | % | p-value | Phi (φ) | % | p-value | Phi (φ) | % | p-value | Phi (φ) | B-M p-value | M-E p-value | B-E p-value |
| **Control** | (55/155) 35.5% | 0.91[c] | 0.007 | (57/154) 37.0% | 0.057[c] | 0.110 | (55/142) 38.7% | **0.017[c]** | 0.141 | 0.50[m] | 0.25[m] | 0.063[m] |
| **Intervention** | (56/155) 36.1% | | | (69/144) 47.9% | | | (76/144) 52.8% | | | **<0.001[m]** | **0.016[m]** | **<0.001[m]** |

[c] = Chi-square test, B = Baseline, M = Midterm, E = Endline, [m] = McNemar Test

Recruiting criteria of CHWs with a minimum of a secondary level of education aligns with the criteria outlined in the revised National Female Community Health Volunteer (FCHV) program strategy of 2019 [14]. According to the strategy, FCHVs are local community women who serve as a bridge between the community and government health facilities [14]. So, we recruited and trained 10 participants considering similar potential and qualifications of future FCHVs.

The knowledge score in our study was similar to a community-based randomized control trial study conducted among women in a semi-urban area of Pokhara Metropolitan City, Nepal [21]. The median knowledge score increased by almost four times in the intervention group after the 3rd week and 6th week follow-up. The average knowledge score was also significantly increased in the control group, although there were significant differences in the knowledge score among the control and intervention groups at all three timepoints of the study. Previous studies in underdeveloped and developing countries have found that health education interventions are effective in enhancing knowledge of cervical cancer among women [21–25]. The increase in cervical cancer knowledge scores observed in both groups may be attributed to the use of health education materials, with more pronounced improvements in the intervention group likely due to the interactive nature of CHW-led sessions.

The national target for cervical cancer screening uptake is 70% by the year 2030 [7]. The cervical cancer screening uptake of the intervention group increased from 36.1% at baseline study to 47.9% in the midterm and ultimately to 52.8% in the endline. The screening uptake in the intervention group was significant across all three time points of the study. There was a minimal increment in cervical cancer screening uptake in the control group, with 35.5% in the baseline to 38.7% in the endline. Several studies on assessing the effectiveness of health education intervention on cervical cancer screening uptake have shown similar findings [19,22,26–28]. The change in cervical cancer screening uptake among the intervention group is lower than that of the experimental study conducted in semi-urban areas of the Pokhara Metropolitan, Nepal [21]. Although cervical cancer screening uptake (CCSU) significantly improved in the intervention group, it still fell short of the national target by approximately 18%. This shortfall may be partly attributed to the limited intervention duration and frequency of CHW mobilization in the current study. The timeline for our study was six weeks, with CHWs mobilized only once, immediately following the baseline survey. In contrast, a previous study conducted over one year engaged Female Community Health Volunteers (FCHVs) on three separate occasions [21]. The difference in CCSU

outcomes between the two studies may therefore be influenced by both the frequency of health education sessions and the overall duration of intervention exposure.

There were significant differences in perceived susceptibility, perceived severity, and behavioral intention regarding cervical cancer screening between the intervention and control groups at both the midterm and endline assessments. In the intervention group, the median scores for perceived susceptibility and perceived severity increased significantly from baseline to midterm, indicating that the health education intervention effectively raised participants' awareness of personal risk and the seriousness of cervical cancer. These findings are consistent with a quasi-experimental study among marginalized women in western Iran, which reported significant improvements in perceived susceptibility, perceived severity, and behavioral intention following educational intervention to prevent cervical cancer [29]. Similarly, studies from Greece and Egypt demonstrated that health education sessions significantly improved the two constructs of the health belief model [30,31]. However, contrasting findings were reported in studies from Ghana and the United States, where perceived susceptibility decreased after the intervention [25,32]. This may be due to the correction of misconceptions or because participants felt they were at lower risk after learning more about cervical cancer and its etiology. Overall, these comparisons suggest that while education generally improves perceived risk and intention, the direction of change in perceived susceptibility may vary based on content delivery, cultural context, and existing beliefs. Our results reinforce the utility of the Health Belief Model in understanding how structured, CHW-led interventions can modify key psychological constructs that influence screening behavior.

An important consideration when interpreting the results of this study is the difference in resources required for the two interventions. The CHW-led approach was resource-intensive, involving the recruitment and training of community health workers, preparation of flip charts and brochures, and supervision of home visits. In contrast, the CIC-led intervention required minimal resources, limited to the placement of posters in public locations. The higher effectiveness of the CHW-led intervention may therefore be partly attributable to its greater resource intensity, including personalized education and interactive materials, whereas the CIC-led intervention relied only on static posters, which may have been less engaging and harder to comprehend for participants with limited literacy.

This study had several notable strengths. It employed a quasi-experimental design with three time-point assessments (baseline, midterm, and endline), which allowed for the evaluation of both short-term effects and changes over time. The use of community health workers (CHWs) to deliver culturally relevant health education through flipcharts and brochures ensured that the intervention was both accessible and contextually appropriate for women living in slum areas. Furthermore, the study measured multiple outcomes, including knowledge, perceptions (perceived susceptibility and severity), behavioral intention, and actual screening uptake anchored in established theoretical frameworks such as the Health Belief Model (HBM) and the Theory of Planned Behavior (TPB), which enhanced the conceptual rigor and interpretation of results.

Despite its strengths, the study also had some limitations. A key limitation of this study is the comparison of two different interventions in separate populations that differed in ethnicity, major source of health information and income. Consequently, differences in outcomes may not be attributed solely to the interventions, as part of the observed effects may reflect inherent differences between the populations. Randomization was not feasible due to logistical and ethical considerations. Therefore, residual confounding may exist, and these findings should be interpreted with caution. Future studies employing randomized or crossover designs would strengthen causal inferences regarding the effectiveness of these interventions. Additionally, the relatively short duration of the intervention and the one-shot mobilization of community health workers may have constrained the potential for longer-term behavior change. Furthermore, this study used visual inspection with acetic acid (VIA) as the screening method, whereas the WHO 90-70-90 strategy recommends HPV-based screening. VIA was chosen due to feasibility and resource constrainment in the study setting; therefore, comparisons with the WHO target should be interpreted cautiously, as the screening method used deviates from current official recommendations.

## Conclusion

Although both the Community Intervention Channel (CIC)-led and Community Health Worker (CHW)-led intervention were effective in improving cervical cancer related knowledge, perceived susceptibility, and perceived severity, only the later brought about the changes in behavioral intention, and thereby increasing cervical cancer screening uptake among women aged 30–60 years in slum areas of Pokhara, incorporating with individual approach. Nonetheless, this didn't meet the national target due to the short intervention period and one-time CHW mobilization.

## Supporting information

**S1 Checklist. CONSORT checklist.**
(DOCX)

**S1 Protocol. Study protocol.**
(DOCX)

**S1 Data. Study data.**
(XLSX)

## Acknowledgments

We sincerely acknowledge the biostatistician of the School of Health and Allied Sciences (SHAS), Pokhara University, Assoc. Prof. Dr. Niranjan Shrestha, for review and comments during proposal and subsequent presentations. We are also indebted to the efforts of the data enumerators Ms. Jyotika Shakya, Ms. Ranjita Kumari Bohara, Ms. Pratiksha Thapa, Ms. Aruna Subedi, and Ms. Shristi Ojha, whose dedication and hard work during data collection were vital to the success of this study. We are deeply grateful to the community health workers from the slum areas of Pokhara Metropolitan for their exceptional support in implementing the intervention, along with the trainer, Mrs. Alisha Thapa. We are grateful to the Gandaki Province Academy of Science and Technology (GPAST) for the financial support that made this research possible.

## Author contributions

**Conceptualization:** Abhishek Sapkota, Chiranjivi Adhikari.

**Data curation:** Abhishek Sapkota.

**Formal analysis:** Abhishek Sapkota.

**Investigation:** Abhishek Sapkota.

**Methodology:** Abhishek Sapkota.

**Project administration:** Abhishek Sapkota.

**Resources:** Abhishek Sapkota.

**Software:** Abhishek Sapkota.

**Supervision:** Chiranjivi Adhikari.

**Validation:** Abhishek Sapkota, Chiranjivi Adhikari.

**Visualization:** Abhishek Sapkota.

**Writing – original draft:** Abhishek Sapkota.

**Writing – review & editing:** Abhishek Sapkota, Chiranjivi Adhikari, Hari Prasad Kaphle, Birkha Bahadur Bist.

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
