## [Decision Letter · Decision Letter 0]

22 Aug 2025

PGPH-D-25-01461

Effectiveness of Health Education Interventions for Cervical Cancer Screening: A Quasi-Experimental Study in Pokhara Metropolitan Slum, Nepal

Dear Dr. Sapkota,

Thank you for submitting your manuscript to PLOS Global Public Health. After careful consideration, we feel that it has merit but does not fully meet PLOS Global Public Health’s publication criteria as it currently stands. Therefore, we invite you to submit a revised version of the manuscript that addresses the points raised during the review process.

This manuscript has been evaluated by 2 reviewers and their concerns are included in full below.

The reviewers raise concerns over the reporting and comparison of inteventions and discussion of the study limitations. In addition, they raise concerns over the statistical approach taken during data analysis.

Could you please revise the manuscript to carefully address the concerns raised?

We look forward to receiving your revised manuscript.

Kind regards,

Jen Edwards

Staff Editor

Journal Requirements:

1. Please amend your online detailed Financial Disclosure statement. This is published with the article. It must therefore be completed in full sentences and contain the exact wording you wish to be published.

a) State the initials, alongside each funding source, of each author to receive each grant, if applicable. For example: "This work was supported by the National Institutes of Health (####### to AM; ###### to CJ) and the National Science Foundation (###### to AM)."

For more information, please go to our submission guidelines:

https://journals.plos.org/globalpublichealth/s/submission-guidelines#loc-financial-disclosure-statement

2. Please ensure that the funders and grant numbers match between the Financial Disclosure field and the Funding Information tab in your submission form. Note that the funders must be provided in the same order in both places as well.

3. In the online submission form, you indicated that “ Data will be made available upon the request by the corresponding author.”.

a) In a public repository,

b) Within the manuscript itself, or

c) Uploaded as supplementary information.

4. Please include a separate legend or caption for Figure 1 in your main manuscript.

5. We have noticed that you have uploaded Supporting Information files, but you have not included a list of legends. Please add a full list of legends for your Supporting Information files before or after the references list.

Additional Editor Comments (if provided):

Reviewers' comments:

Reviewer's Responses to Questions

**Comments to the Author**

1. Does this manuscript meet PLOS Global Public Health’s publication criteria?

Reviewer #1: Yes

Reviewer #2: Partly

2. Has the statistical analysis been performed appropriately and rigorously?

Reviewer #1: No

Reviewer #2: Yes

3. Have the authors made all data underlying the findings in their manuscript fully available (please refer to the Data Availability Statement at the start of the manuscript PDF file)?

Reviewer #1: Yes

Reviewer #2: Yes

4. Is the manuscript presented in an intelligible fashion and written in standard English?

Reviewer #1: Yes

Reviewer #2: Yes

Reviewer #1: Major revision

This appears to be a well designed (but non-randomised) study the results of which would be of considerable interest. Unfortunately, the statistical approach to analysis is far from optimal and mostly inappropriate. Although not immediately statistical, the major shortcoming is that the endpoints used for the evaluations are not defined. Thus, although the authors state (lines 32-33), “The intervention group demonstrated significant improvements in knowledge, perceived susceptibility, perceived severity, behavioral intention, and screening uptake ...” no scale for each outcome measure is defined. Without this information, further review is not useful.

My advice to the authors is to find help from an experienced medical statistician in preparing a revision of this paper

Reviewer #2: Thank You for an interesting paper dealing with an important subject- how to improve participation in cervical cancer screening among underscreened women. And I must congratulate You on the excellent results obtained when Health Care Workers approach the women with informations regarding the importance of screening.

My major concern is the design of the study, where You compare two different interventions in two seperate populations. From the start You show with You analysis that the populations differ significantly on core aspects and consequentlyly it is hard to conclude whether differences in outcomes are only based on interventions or may at least partly be attributed to differences in populations. As design can not be changed I recommend that You discuss this weakness in depth.

You test two different interventions which are very different in use of ressources etc. It is mandatory that the interventions are described in more details. For example - the written material could be inappropriate and explain the lack of effect. Have You performed any testing ahaed of using the intervention?

Moreover, the questionaire must be presented with more details to provide the reader with sufficient informations for evaluating the results.

Given that the two interventions are very different with regards to use of ressources I recommend that You also descriebe the intervention provided by Health Care Workers in more detail and compare the two interventions in this perspective.

In general I suggest a condensation of the manuscript. Many results are given in both Tabels and repated in the result section - please choose and I suggest to reduce the number of Tables.

Several times You refer to the WHO 90-70-90 strategy which is Your outcome goal. However, the WHO strategy aims to have 70% of younger women screened by HPV whereas You screen by VIA which is no longer recommended by the WHO. I am aware that VIA is trhe only possibility iin some countries by I think Youn should emphasize the deviation from the official recommendations.

**Do you want your identity to be public for this peer review?** For information about this choice, including consent withdrawal, please see our Privacy Policy

Reviewer #1: No

Reviewer #2: No

---

## [Decision Letter · Decision Letter 1]

14 Nov 2025

PGPH-D-25-01461R1

Effectiveness of Health Education Interventions for Cervical Cancer Screening: A Quasi-Experimental Study in Pokhara Metropolitan Slum, Nepal

Dear Dr. Sapkota,

Thank you for submitting your manuscript to PLOS Global Public Health. After careful consideration, we feel that it has merit but does not fully meet PLOS Global Public Health’s publication criteria as it currently stands. Therefore, we invite you to submit a revised version of the manuscript that addresses the points raised during the review process.

The manuscript has been evaluated by one reviewer, and their comments are available below.

The reviewer has raised concerns with the statistical analysis.

Could you please revise the manuscript to carefully address the concerns raised?

We look forward to receiving your revised manuscript.

Kind regards,

Alejandro Torrado Pacheco, PhD

Staff Editor

Journal Requirements:

Additional Editor Comments (if provided):

Reviewers' comments:

Reviewer's Responses to Questions

**Comments to the Author**

Reviewer #1: (No Response)

publication criteria?

Reviewer #1: Yes

3. Has the statistical analysis been performed appropriately and rigorously?

Reviewer #1: No

4. Have the authors made all data underlying the findings in their manuscript fully available (please refer to the Data Availability Statement at the start of the manuscript PDF file)?

Reviewer #1: Yes

5. Is the manuscript presented in an intelligible fashion and written in standard English?

Reviewer #1: Yes

Reviewer #1: Recommedation

Major revision

As indicated before, this appears to be a well-designed (but non-randomised) study the results of which would be of considerable interest. The previous major shortcoming was that the endpoints used for the evaluations were not defined. This has now been rectified.

However, the statistical analysis requires some extensive modification. Some issues are listed below:

1. Table 1 and elsewhere: The IQR is not a very useful format for describing ages groups. Better to quote the median, minimum and maximum values in the two intervention groups. This applies to age and monthly income.

2. Table 1 and elsewhere: P-values should be quoted to two significant figures, for example ‘0.114’ becomes ‘0.11’ and ‘0.845’ becomes ‘0.85’ while ‘0.070’ stays as it is.

3. Table 2: If for example, the endpoint Knowledge cannot be assumed to have a Normal distribution, then median, minimum and maximum of each group should be given, together with the difference between the two medians (this is not a straight forward calculation of median1–median2), and the corresponding 95% confidence interval (CI). A CI is easier to interpret than an effect size (lines 234-237). These calculations require appropriate statistical software, but I don’t know if this is available on SPSS. On the other hand, if the data can be assumed approximately Normal means can be used and things become a lot easier.

4. Table 3: This overlaps with the information in Table 1. The latter compares between interventions the former within interventions. These two would be better combined in a trend analysis. For example, from Table 1 the mean value of the three Intervention medians for Knowledge is (2 + 7.5 +8)/3 = 5.83 and that for the Control (3 + 5 +5)/3 = 4.33. So, the trend upwards over time is greater in the Intervention and this is really what is required for Table 3. Strictly this type of calculation should be made for each subject and then averaged within each group and then compared between two groups.

**Do you want your identity to be public for this peer review?** For information about this choice, including consent withdrawal, please see our Privacy Policy

Reviewer #1: No

---

## [Decision Letter · Decision Letter 2]

18 Dec 2025

PGPH-D-25-01461R2

Effectiveness of Health Education Interventions for Cervical Cancer Screening: A Quasi-Experimental Study in Pokhara Metropolitan Slum, Nepal

Dear Dr. Sapkota,

Thank you for submitting your manuscript to PLOS Global Public Health. After careful consideration, we feel that it has merit but does not fully meet PLOS Global Public Health’s publication criteria as it currently stands. Therefore, we invite you to submit a revised version of the manuscript that addresses the points raised during the review process.

We look forward to receiving your revised manuscript.

Kind regards,

Nnodimele Onuigbo Atulomah, PhD

Academic Editor

Journal Requirements:

Additional Editor Comments (if provided):

It has been observed that there is need to make some minor but important corrections in how p-values are presented in the manuscript submitted. Kindly attend to this recommendation for revision before a final decision can be made about the manuscript.

Reviewers' comments:

Reviewer's Responses to Questions

**Comments to the Author**

Reviewer #1: (No Response)

publication criteria?

Reviewer #1: Yes

3. Has the statistical analysis been performed appropriately and rigorously?

Reviewer #1: No

4. Have the authors made all data underlying the findings in their manuscript fully available (please refer to the Data Availability Statement at the start of the manuscript PDF file)?

Reviewer #1: Yes

5. Is the manuscript presented in an intelligible fashion and written in standard English?

Reviewer #1: Yes

Reviewer #1: Minor revision

The authors have made important changes to the manuscript. One mino rbut important problem remains.

My previous review stated:

“2. Table 1 and elsewhere: P-values should be quoted to two significant figures, for example ‘0.114’ becomes ‘0.11’ and ‘0.845’ becomes ‘0.85’ while ‘0.070’ stays as it is.”

This does not mean 2 decimal places. So the many places where p < 0.01 are quoted should be replaced by p= 0.01x where x is the second significant figure. For example, p < 0.01 could imply that p = 0.009 or even 0.005 which should then be reported as 0.009X and 0.005X

**Do you want your identity to be public for this peer review?** For information about this choice, including consent withdrawal, please see our Privacy Policy

Reviewer #1: No

---

## [Decision Letter · Decision Letter 3]

6 Jan 2026

Effectiveness of Health Education Interventions for Cervical Cancer Screening: A Quasi-Experimental Study in Pokhara Metropolitan Slum, Nepal

PGPH-D-25-01461R3

Dear Dr. Sapkota,

We are pleased to inform you that your manuscript 'Effectiveness of Health Education Interventions for Cervical Cancer Screening: A Quasi-Experimental Study in Pokhara Metropolitan Slum, Nepal' has been provisionally accepted for publication in PLOS Global Public Health.

Best regards,

Nnodimele Onuigbo Atulomah, PhD

Academic Editor

Having satisfied the reviewers at all stages of the review process, I am recommending that the manuscript be considered for publication.

Reviewer Comments (if any, and for reference):

Reviewer's Responses to Questions

**Comments to the Author**

Reviewer #1: All comments have been addressed

publication criteria?

Reviewer #1: Yes

3. Has the statistical analysis been performed appropriately and rigorously?

Reviewer #1: Yes

4. Have the authors made all data underlying the findings in their manuscript fully available (please refer to the Data Availability Statement at the start of the manuscript PDF file)?

Reviewer #1: Yes

5. Is the manuscript presented in an intelligible fashion and written in standard English?

Reviewer #1: Yes

Reviewer #1: (No Response)

**Do you want your identity to be public for this peer review?** For information about this choice, including consent withdrawal, please see our Privacy Policy

Reviewer #1: No
